# The Metabolic Pathways of Yeast and Acetic Acid Bacteria During Fruit Vinegar Fermentation and Their Influence on Flavor Development

**DOI:** 10.3390/microorganisms13030477

**Published:** 2025-02-21

**Authors:** Yinggang Ge, Yifei Wu, Aihemaitijiang Aihaiti, Liang Wang, Yu Wang, Jun Xing, Min Zhu, Jingyang Hong

**Affiliations:** School of Life Science and Technology, Xinjiang University, Urumqi 830017, China; 107552303723@stu.xju.edu.cn (Y.G.); 107552303738@stu.xju.edu.cn (Y.W.); aa2386935884@xju.edu.cn (A.A.); wl1390593786@163.com (L.W.); wangyu170222@163.com (Y.W.); 001319@xju.edu.cn (J.X.)

**Keywords:** fruit vinegar, microorganism, flavor components, metabolic pathway

## Abstract

Fruit vinegar is a beverage derived from fruits or fruit processing by-products through microbial fermentation. This vinegar possesses a distinctive flavor profile and contains bioactive compounds. It is typically produced using liquid fermentation technology. As consumer demand for the flavor quality of fruit vinegar has increased, precise control over flavor compounds has become crucial for enhancing the quality of fermentation products. Vinegar contains numerous characteristic flavor compounds, including esters, aldehydes, alcohols, and organic acids. These unique flavors primarily result from the accumulation of flavor compounds generated by different raw materials and microorganisms during fermentation. Specifically, yeast and acetobacter promote the formation of distinct fruit vinegar flavors by facilitating the breakdown of carbohydrates, amino acids, and proteins in fruits, as well as the redox and esterification reactions involving alcohols. This paper reviews the metabolic pathways of yeast and acetic acid bacteria during fruit vinegar fermentation and discusses key volatile compounds that influence the flavor of fruit vinegar and their potential relationships, providing theoretical support for regulating flavor quality.

## 1. Introduction

With the continuous growth of the global population and increasing attention to dietary health, fruits have assumed an increasingly significant role as a vital source of nutrition in global agricultural production. According to FAO statistics, global fruit production increased from 601 million tons in 2002 to 933 million tons in 2022 (https://www.fao.org/faostat/zh/#data/QCL accessed on 30 December 2024). However, between 25% and 80% of fruits lose their edible value annually during transportation and storage due to their susceptibility to decay [1]. Fruit vinegar, as the name suggests, is a flavorful condiment produced through either single-stage fermentation (acetic acid fermentation) or sequential fermentation (alcohol fermentation followed by acetic acid fermentation), using fresh fruits or fruit processing by-products as raw materials (Figure 1) [2,3]. As an important method of fruit processing, fruit vinegar production offers several key advantages: low raw material requirements, a relatively short fermentation period, high biological activity, and a more diverse flavor profile [4]. During the fermentation process of fruit vinegar, not only are beneficial compounds such as phenols and flavonoids retained from the original fruit, but there is also an increase in volatile aroma components, including esters, ketones, alcohols, and terpenes [5,6,7,8]. Flavor is a critical criterion for evaluating the quality of vinegar. The primary sources of flavor can be attributed to four key factors: (1) the aroma and flavor compounds present in the raw materials; (2) the organic compounds produced by yeast during alcohol fermentation; (3) the further degradation of macromolecular substances during acetic acid fermentation; and (4) the chemical reactions between small molecules during the aging process [9]. From 2014 to 2023, a total of 759 articles were retrieved from the Web of Science database using the themes “fruit vinegar” and “flavor”. The number of articles has exhibited a consistent annual increase. By 2023, the number of articles had grown by approximately fivefold compared to 2014. (Figure 2).

During the initial fermentation stage of vinegar production, yeast serves as the primary fermentative agent. It metabolizes the sugars present in the raw materials to produce alcohol. Concurrently, this process generates flavor compounds such as esters, aldehydes, and higher alcohols, which form the foundation for the characteristic aroma and taste of fruit vinegar [10]. During the initial fermentation stage of vinegar production, yeast serves as the primary fermentative agent. It metabolizes the sugars present in the raw materials to produce alcohol. Simultaneously, this process generates flavor compounds such as esters, aldehydes, and higher alcohols, which form the foundation for the characteristic aroma and taste of fruit vinegar [11,12].

Building upon alcohol fermentation, acetic acid fermentation is a process in which acetic acid is produced using alcohol and sugars as substrates. During this process, acetobacter utilizes carbon sources to generate significant amounts of organic acids and degrades proteins to produce a variety of amino acids. Subsequently, some amino acids and fatty acids undergo decomposition via the tricarboxylic acid (TCA) cycle and oxidative catabolism, leading to the formation of small molecular components such as aldehydes and ketones [13]. These substances are also the primary contributors to the volatile components in vinegar. In addition to the volatile flavor compounds, fruit vinegar contains amino acids, malic acid, citric acid, and other components that contribute to its taste profile. These substances are primarily derived from the metabolic activities of fermentation microorganisms and the enzymatic hydrolysis of proteins, lipids, and other substrates by extracellular enzymes [12]. The objective of this review is to elucidate the metabolic processes of yeast and acetic acid bacteria, as well as the mechanisms underlying the formation of flavor compounds during fruit vinegar fermentation.

In recent years, as research into the fruit vinegar fermentation process has deepened, there has been growing recognition of the critical roles played by yeast and acetic acid bacteria in this process. The objective of this review is to elucidate the metabolic processes of yeast and acetic acid bacteria, as well as the formation of flavor compounds during fruit vinegar fermentation.

## 2. Vinegar Fermentation Process

Vinegar is a fermented food product composed of an aqueous solution of acetic acid and various other components. Despite the diversity of its raw materials and production processes, vinegar has been widely utilized as a food condiment across different cultures and cuisines worldwide [14]. The production process of vinegar involves two primary fermentation stages. Initially, the sugars in the raw materials undergo alcohol fermentation catalyzed by yeast to produce ethanol. Subsequently, ethanol is oxidized to acetic acid through the enzymatic action of acetic acid bacteria, which are key microorganisms in this process [15,16]. Furthermore, raw materials rich in monosaccharides, disaccharides, starches, and complex carbohydrates—such as the juices of vegetables and fruits like grapes, apples, and tomatoes—require a saccharification and enzymatic hydrolysis step prior to alcoholic fermentation to convert these carbohydrates into fermentable sugars [17]. The selection of appropriate bacteria is crucial for ensuring the final quality of fruit vinegar. During the alcohol fermentation stage, yeasts with high alcohol tolerance are required to meet the fermentation requirements for ethanol production. In the acetic acid fermentation stage, it is essential to screen for acetic acid bacteria that exhibit dual tolerance to both alcohol and acetic acid. Moreover, the unique physiological and metabolic characteristics of microorganisms lead to variations in metabolites, which can significantly influence the flavor and aroma profiles of the final product under different raw material conditions (Table 1).

### 2.1. Effect of Yeast Strains on the Quality of Fruit Vinegar

The existing literature indicates that during the processing of fruit vinegar, commercially available Saccharomyces cerevisiae is primarily used for fermentation. In the alcohol fermentation stage, yeast contributes to flavor development through the volatilization of ethanol and the formation of yeast-derived metabolites. Specifically, the volatilization of ethanol imparts an alcoholic aroma to the fruit vinegar, while metabolites such as esters, aldehydes, and higher alcohols contribute to its unique flavor profile, enhancing the overall sensory characteristics of the final product [37]. For instance, ester compounds such as ethyl acetate and isoamyl acetate impart a floral and fruity aroma to vinegar, whereas aldehydes like acetaldehyde and diacetyl contribute a fruity and wine-like fragrance. Notably, different microorganisms produce distinct fermentation products, which in turn influence the nuanced differences in flavor profiles. Through the research on persimmon vinegar, it was discovered that pyrazine compounds, unique fermentation products of *S. delbrueckii* (DSM 70483), contribute to its distinctive baking properties and nutty aroma, thereby enhancing the overall sensory characteristics of the final product [32]. *Pichia manshurica,* isolated from Daqu, was utilized in the fermentation process of Shanxi aged vinegar. This strain significantly increased the yield of organic acids and esters, thereby enhancing the vinegar’s floral and fruity aroma. Specifically, the elevated levels of these compounds contribute to a more complex and desirable flavor profile [38]. Through high-throughput sequencing and a bioinformatics analysis, it was determined that compounds such as ethyl lactate, methyl acetate, betaine, aconitic acid, inositol, and L-isoleucine in persimmon vinegar exhibited a significant positive correlation with the abundance of *Wickerhamomyces anomalus* [23]. Other studies have demonstrated that the lactic acid content in persimmon vinegar is primarily associated with *Dekkera*, *Pichia*, and *Candida* species. The levels of leucine and lysine were predominantly correlated with *Saccharomyces cerevisiae* and *Hanseniaspora* [12].

### 2.2. Effect of Acetic Acid Strains on the Quality of Fruit Vinegar

The selection of acetic acid bacteria primarily involves isolating strains from spoiled raw materials during the processing of fruit vinegar. Ideal strains should exhibit an efficient utilization of raw materials, a high tolerance to alcohol and acetic acid, and a short fermentation cycle. The methods for isolation and identification are as follows: First, a preliminary screening is conducted based on colony morphology and basic metabolic characteristics to identify candidate strains. Second, a detailed characterization and a metabolic analysis are performed to further evaluate their physiological and biochemical properties. Finally, gene phylogenetic trees are constructed through gene library alignment to accurately determine specific genera and subspecies [33,39,40]. During the liquid fermentation of fruit vinegar, acetic acid bacteria primarily produce acetic acid via the sugar metabolism pathway. Acetic acid is a key component of fruit vinegar and plays a crucial role in determining its acidity. Specifically, acetic acid bacteria metabolize ethanol, which is generated by yeast during alcoholic fermentation, into acetic acid through a series of enzymatic reactions [41]. As fermentation progresses, the acetic acid produced by acetic acid bacteria increases, leading to a corresponding rise in the acidity of the fruit vinegar. Additionally, acetobacter generates various flavor compounds, such as esters, aldehydes, and phenols, during the fermentation process. These compounds play a crucial role in determining the aroma and color of fruit vinegar [13]. For instance, ester compounds impart a distinctive fruity aroma to fruit vinegar, whereas aldehydes and phenolic compounds influence its color [42,43]. Different acetic acid bacteria exert distinct effects on the final quality of a vinegar. Studies have shown that fermentation by *Acetobacter pasteurianus* (NBRC 3284) reduces the levels of undesirable flavor compounds, such as allyl cyanide and sulfides (dimethyl sulfide, dimethyl disulfide, and dimethyl trisulfide), in cabbage. However, this fermentation has no significant impact on the unique aroma components, such as crotonic acid and 3-hexen-1-ol [34]. When comparing the effects of three strains of acetic acid bacteria on the quality of black tea vinegar, *Acetobacter* sp. (GDMCC1.152) exhibited significantly higher concentrations of volatile flavor components, such as (Z,E)-farnesol, linalyl formate, ethyl 3-hydroxybutyrate, α-santalol, 2,3-epoxycitral, α-bisabolol oxide B, ionone, and dihydroactinidiolide, compared to the other two acetobacter strains. Additionally, this strain produced black tea vinegar with superior bioactive components, including tea polyphenols (TPs), thearubigins (TRs), and theabrownins (TBs), compared to the vinegar fermented by the other two strains [3]. Researchers utilized three strains of acetic acid bacteria—*Acetobacter pasteurianus* KACC16934 (AP), *Acetobacter malorum* V5-7 (AM), and *Gluconoacetobacter entanii* RDAF-S (GE)—to evaluate their effects on the quality of Rhus verniciflua vinegar. The analysis revealed that the main free amino acids in the vinegar fermented by AP were arginine (277.74 μg/mL), tyrosine (160.17 μg/mL), and alanine (158.95 μg/mL). For the vinegar fermented by AM, the primary free amino acids were arginine (277.74 μg/mL), alanine (181.90 μg/mL), and proline (172.85 μg/mL). In contrast, the vinegar fermented by GE contained arginine (278.28 μg/mL) and alanine (273.67 μg/mL) [44].

## 3. Microbial Metabolism in Vinegar Production

Liquid-fermented fruit vinegar is a food product characterized by its unique flavor, with yeast and acetic acid bacteria playing crucial roles as microorganisms. The metabolic activities of these microorganisms directly influence the quality and flavor profile of fruit vinegar.

### 3.1. Sugar Metabolism of Saccharomyces Cerevisiae in Fruit Vinegar Fermentation

Yeast serves as a crucial microorganism during the initial stage of liquid fermentation in fruit vinegar production. Its primary function is to convert sugars into ethanol and carbon dioxide, even under aerobic conditions [45,46]. The ethanol produced by yeast metabolism serves as a critical precursor for acetic acid formation. Higher ethanol concentrations generally correlate with increased acidity in fruit vinegar. During fermentation, carbon dioxide is gradually released, causing the formation of bubbles that can enhance the taste and freshness of the vinegar. Carbon dioxide is not only a by-product of yeast fermentation but also an important signaling molecule that regulates yeast metabolism and stress responses. Researchers have summarized the role of carbon dioxide in yeast fermentation as follows: carbon dioxide can modulate the mitogen-activated protein kinase (MAPK) signaling pathway genes, reducing glycerol and acetic acid production while increasing ethanol production. It promotes the up-regulation of glycogen and trehalose synthesis-related genes (GSY1, GSY2, GPH1, UGP1, GLC3, GDB1, TSL1, and TPS2), leading to increased carbohydrate synthesis and enhanced adaptability to environmental stresses. Additionally, carbon dioxide enhances the expression of high-affinity glucose transporter genes (HXT6 and HXT7), promoting glucose uptake in yeast (Figure 3) [47]. The mechanism of sugar metabolism in yeast is intricate. In most commercial fermentations, insufficient nitrogen sources often lead to fermentation metabolism and glycolysis becoming predominant factors. Yeast achieves the preferential utilization of glucose through complex genetic programs, a phenomenon known as glucose repression. This process facilitates the reduction of pyruvate to ethanol via fermentation, promoting rapid alcohol production, minimizing the lag phase, and inhibiting the growth of competing microorganisms [48,49]. In the early stages of yeast fermentation, yeast cells are exposed to high sugar concentrations, leading to osmotic and oxidative stress. This environment results in significant alterations in yeast metabolism and an increase in the overall carbon metabolic flux. During this initial phase, trehalose and glycerol metabolism predominate, while the pentose phosphate pathway remains relatively inactive. The tricarboxylic acid (TCA) cycle exhibits a fluctuating pattern, initially increasing, then decreasing, and subsequently rising again. Meanwhile, glycolytic activity continues to intensify [50]. The transcription factor Gln3 plays a crucial role in regulating nitrogen and carbon metabolisms during yeast fermentation. The nitrogen-induced TORC1 complex, along with the regulation of carbon catabolite repression, activates the kinase Snf1 to ensure proper metabolic activity in yeast [51,52].

Yeast is a eukaryotic microorganism that is extensively utilized in the production of fruit wine, fruit vinegar, and jams. During the fermentation process, yeast converts sugars, amino acids, and esters into ethanol, serving as the primary organism responsible for alcohol compound synthesis [53]. Alcohol compounds, as volatile substances, are prevalent in a wide array of food products and exhibit a relatively low sensory threshold, contributing to the development of unique flavor profiles. In addition to ethanol, the alcohol compounds in fruit vinegar include isopentanol, which imparts a fruity aroma; phenylethanol, which provides a floral scent; and isobutanol [54,55,56]. In fruit wine, 3-methyl-1-butanol, 2-methylpropanol, 2-phenylethanol, and benzyl alcohol are key volatile compounds that contribute to the wine’s flavor and fruity aroma, in addition to ethanol [20,21]. Higher alcohols undergo esterification reactions to produce ester compounds under the catalysis of alcohol acetyltransferase (AAT). In yeast cells, acetic acid first binds to coenzyme A (CoA) to form acetyl-CoA. Subsequently, AAT catalyzes the esterification of acetyl-CoA with higher alcohols, such as isoamyl alcohol and active amyl alcohol. This reaction transfers the acetyl group from acetyl-CoA to the hydroxyl group of higher alcohols, forming corresponding ester compounds. These esters are then released into the wine, contributing to its aroma profile [22]. In the fermentation process, while alcohols play a crucial role in determining the overall flavor profile, other low-abundance primary metabolites also significantly influence flavor characteristics. These metabolites, including glycerol, succinate, acetate, and lactate, not only enhance the complexity of fermentation products but are also closely associated with yeast metabolic capacity [54,57,58]. For instance, glycerol and succinate levels exhibit negative correlations with ethanol content; acetate levels are negatively correlated with yeast domestication degree, and lactic acid levels show positive correlations with yeast glycolysis activity [59].

### 3.2. Acetobacter Metabolism in Vinegar Production

Acetobacter is a significant acid-producing bacterium in the fermentation industry and is extensively utilized in food brewing. These bacteria oxidize ethanol to acetic acid, thereby increasing the acidity of fruit vinegar. Acetic acid bacteria are classified into two main categories: the first category is acetic acid bacteria, whose primary function is to oxidize ethanol to acetic acid, serving as the core microbial strain in fruit vinegar production; the second category is gluconic acid bacteria, which primarily oxidize glucose to gluconic acid. Common acetobacter species include acetobacter aceti, acetobacter schuezenbachii, acetobacter pasteurianus, etc. Acetic acid bacteria possess robust redox capabilities, enabling them to convert ethanol and glucose into acetic acid and gluconic acid. Additionally, they can oxidize various alcohols and sugars to produce a range of organic acids, including butyric acid, pyruvic acid, succinic acid, lactic acid, and others. Furthermore, these bacteria can oxidize glycerol to diketone and mannitol to fructose [60]. ADH (pyrroloquinoline quinone-dependent alcohol dehydrogenase) is a key enzyme in the ethanol oxidation respiratory chain of acetic acid bacteria. In acetobacter pasteurianus, PQQ-ADH is a multi-subunit complex comprising three subunits: Subunit I (adhA) encodes the dehydrogenase subunit, which contains PQQ and one molecule of heme c; Subunit II (adhB) encodes the cytochrome c subunit, which contains three heme C molecules and is involved in membrane binding and ubiquinone reduction; and Subunit III (adhS) encodes the smallest subunit, which appears not to be essential for ethanol oxidation [61]. The first step in the ethanol oxidation process involves the conversion of ethanol to acetaldehyde. This reaction is catalyzed by pyrroloquinoline quinone-dependent alcohol dehydrogenase (PQQ-ADH), which oxidizes ethanol to acetaldehyde, resulting in the reduction of PQQ to PQQH2. The second step entails the further oxidation of acetaldehyde to acetic acid (CH3COOH). This reaction is mediated by aldehyde dehydrogenase (ALDH), leading to the reduction of NAD+ to NADH. Notably, the enzymatic activity in this ethanol oxidation process is not regulated at the transcriptional level [62,63]. Acetic acid, the final product of ethanol oxidation, can enter the TCA cycle directly in the form of acetyl-CoA. Acetic acid is first activated by acetyl-CoA synthetase, which catalyzes the reaction between acetic acid and coenzyme A (CoA) to produce acetyl-CoA and an inorganic phosphate (Pi). Acetyl-CoA serves as a direct substrate for the TCA cycle. In the TCA cycle, acetyl-CoA condenses with oxaloacetate to form a citrate, a reaction catalyzed by citrate synthase. Subsequently, the citrate undergoes a series of redox reactions, ultimately regenerating oxaloacetate while producing ATP, NADH, and FADH2 [64]. Simultaneously, bacteria possess a bifunctional enzyme, acetaldehyde-alcohol dehydrogenase (AdhE), which exhibits both aldehyde dehydrogenase (AlDH) and alcohol dehydrogenase (ADH) activities. Under anaerobic conditions, AdhE can convert acetyl-CoA to ethanol. This process not only generates energy but also maintains the intracellular NAD+/NADH balance by oxidizing NADH and regenerating NAD+, which is essential for sustaining bacterial glycolysis [65]. Following these reactions, acetobacter can effectively mitigate ethanol stress in the environment while maintaining its carbon metabolic capacity.

In addition to acetic acid and some flavor components, some bioactive polysaccharides are also produced during the metabolism of acetobacter. It encompasses both homopolysaccharides, such as levan and bacterial cellulose, and heteropolysaccharides, including xylan and glucan acetate. Notably, levan-type fructan and bacterial cellulose are the most widely utilized polysaccharides in industrial applications [66]. The natural homopolysaccharide, levan-type fructan, contains β-(2, 6) glycosidic bonds in its main chain and β-(2, 1) glycosidic bonds in its branch chains. This polysaccharide is primarily synthesized through the polymerization of sucrose catalyzed by levansucrase, an enzyme of microbial origin. During transfructosylation reactions, levansucrase transfers fructose residues from sucrose to the growing levan chains in the extracellular matrix, thereby synthesizing levan [66]. Bacterial cellulose consists of multiple glucose monomers linked by β-(1, 4) glucan chains. The synthesis of bacterial cellulose begins with the conversion of acetyl coenzyme A into uridine diphosphate glucose (UDP-Glucose), catalyzed by the Cellulose Synthase Complex (CSC). Subsequently, UDP-Glucose is polymerized into β-1, 4-glucan chains through the action of the cellulose synthase system. These newly formed glucan chains are then transported outside the cell by the cellulose synthase complex and self-assemble into microfibrillar structures on the cell surface or within the extracellular matrix. As more glucan chains are secreted and assembled, they intertwine to form a three-dimensional network structure, ultimately resulting in bacterial cellulose membranes [67].

## 4. Flavor Formation in Vinegar

Fruit vinegar combines the flavors of vinegar and fruit, offering excellent taste and rich nutritional content. The flavor compounds in fruit vinegar primarily consist of organic acids, alcohols, esters, phenols, and other volatile substances. The interaction of these compounds determines the distinctive flavor profile of fruit vinegar. Microorganisms produce these flavor compounds through various metabolic pathways, including amino acid metabolism, fatty acid metabolism, carbohydrate metabolism, and nucleotide metabolism. These pathways generate primary metabolites such as amino acids, short-chain peptides, fatty acids, and nucleotides. Subsequently, microorganisms further metabolize these primary metabolites into secondary metabolites, which are eventually decomposed into volatile compounds that contribute to the characteristic aroma and taste of fruit vinegar [68]. Yeast and acetobacter exert distinct influences on the production and accumulation of flavor compounds during fruit vinegar fermentation (Figure 4).

### 4.1. Non-Volatile Components

#### 4.1.1. Saccharides

Sugars serve as the primary energy source for microbial fermentation. Acetylphosphoric acid and pyruvate, which are carbon metabolites produced during the early stages of fermentation, act as crucial intermediates in the synthesis of acetic acid and lactic acid [69]. Prior to fermentation, sucrose is typically supplemented to achieve a total sugar content of over 15 °Brix in the fermentation medium, ensuring optimal energy metabolism for yeast and acetic acid bacteria. As fermentation progresses, sugars are continuously consumed, leading to a reduction in the total sugar content to below 6 °Brix, marking the end of alcoholic fermentation and the onset of acetic acid fermentation [70,71]. During the fermentation process, microorganisms secrete carbohydrate-active enzymes (CAZymes) to degrade insoluble macromolecular polysaccharides into smaller, more soluble molecules such as oligosaccharides and monosaccharides, thereby meeting their requirements for growth and proliferation [72]. Emerging evidence indicates that microbial metabolism and nutrient oxidation during fermentation can generate reactive oxygen species (ROS) and reactive nitrogen species (RNS), leading to the fragmentation of polysaccharides, which may alter their molecular functions [73]. It has been observed that yeast and acetobacter aceti can produce functional polysaccharides during fermentation, primarily via the Wzx/Wzy-dependent pathway, the ABC transporter-dependent pathway, the synthase-dependent pathway, and extracellular synthesis mechanisms [74]. The polysaccharides produced during fermentation can impart a rich flavor to food [75]. For example, soluble cell wall carbohydrates and phenolic compounds derived from grapes and yeasts play a significant and complementary role in influencing the taste of wine [76]. Millet wine polysaccharides can form complexes with ethyl acetate, acetic acid, 2,3-butanediol, and γ-butyrolactone through van der Waals forces and hydrogen bonds. This interaction inhibits the release of herbal, fruity, and alcoholic aromas while enhancing the sweetness and rice aroma [77].

Recent studies have demonstrated that polysaccharides produced through fermentation exhibit various functional activities [78,79]. The crude KPV-I polysaccharide fraction from persimmon vinegar, primarily comprising arabinose, mannose, galactose, rhamnose, and galacturonic acid, exhibits a potent capacity to induce macrophage-stimulating factor production [80]; the polysaccharides, including mannose, arabinose, rhamnose, galactose, and a lesser amount of glucose, in longan vinegar can promote cytokine secretion by activating the MAPK and PI3K/Akt signaling pathways [74]. Apple vinegar crude polysaccharides, composed of 38.2% mannose, 19.1% galactose, and 14.3% glucose, stimulate macrophages and RAW 264.7 cells to produce a variety of cytokines, including interleukin, tumor necrosis factor, and nitric oxide [81]. The aforementioned studies have demonstrated that, despite constituting a relatively small proportion of vinegar, polysaccharides play a significant role in enhancing immunity and modulating flavor.

#### 4.1.2. Amino Acids

During the vinegar formation process, the concentration of amino acids typically follows an initial increase followed by a subsequent decrease [82]. The free amino acids in vinegar are primarily derived from the microbial decomposition of proteins in the raw materials and through microbial anabolism. Microorganisms utilize intermediates from the tricarboxylic acid (TCA) cycle, such as pyruvate and phosphoenolpyruvate, to further synthesize various free amino acids via transamination, reductive amination, decarboxylation, and other metabolic reactions [83]. For example, glutamic acid is aminated from α-ketoglutaric acid via glutamate dehydrogenase, and alanine is transaminated from pyruvate by pyruvate transaminase. Free amino acids are among the primary metabolites in vinegar fermentation. They not only serve as a source of human nutrition but also influence product quality and consumer purchasing intentions. Additionally, free amino acids can react with sugars in the raw materials to produce small-molecule flavor compounds and pigments [84]. Glutamic acid is the most abundant free amino acid in fruit vinegar, imparting a distinct umami flavor and contributing to its complex taste profile. Its concentration increases with an extended aging time [85]. Studies have demonstrated that the amino acid profiles of wine and grape vinegar differ primarily in citrulline, glutamine, and tryptophan, while the remaining amino acids fall within similar ranges [86]. Glutamine and alanine contribute umami and sweetness to the flavor of vinegar, resulting in a high taste activity value (TAV). Amino acids not only enhance the flavor of vinegar but also exhibit significant biological activities. Specifically, the valine, leucine, and isoleucine in vinegar can help mitigate diet-induced obesity and hyperglycemia while increasing insulin sensitivity. Additionally, leucine stimulates protein synthesis, leading to increased energy expenditure and fatty acid oxidation [87]. In addition, γ-aminobutyric acid (GABA), a non-protein amino acid produced during fermentation, can enhance blood circulation and cellular metabolism in the brain. It also exhibits significant sleep-promoting, anti-anxiety, and sedative effects [85]. In summary, the amino acids present in vinegar not only influence its sensory attributes but also serve as key bioactive components that promote human health.

### 4.2. Volatile Flavor Components

#### 4.2.1. Organic Acids

Organic acids are the primary functional and flavor components of vinegar, with the total acid content in fruit vinegar being higher than that in grain vinegar (such as sorghum, rice, and wheat). The organic acids in vinegar consist of volatile organic acids—primarily produced during the fermentation stage, including acetic acid, formic acid, propionic acid, butyric acid, and quinic acid—and non-volatile organic acids, such as lactic acid, malic acid, pyroglutamic acid, citric acid, and succinic acid, which originate from the raw materials and account for 0.5% to 2% of the total acid content in fruit vinegar [7]. Simultaneously, the hydrolysis of plant proteins during fermentation releases free amino acids that contribute to taste, as described in the previous chapter. In addition to imparting acidity, volatile organic acids also provide essential flavor notes to vinegar. The malic acid and citric acid from the raw materials not only impart a fruity and complex flavor to the vinegar but also help neutralize the sharpness of the acetic acid, resulting in a softer and more delicate vinegar [82]. Lactic acid is produced by yeast during the alcohol fermentation stage. Although its sourness is relatively mild, it adds a soft dimension to the flavor profile of vinegar. The concentration of lactic acid decreases as acetic acid fermentation progresses [88]. Under aerobic conditions, acetic acid bacteria convert alcohol into acetic acid. Acetic acid is the predominant organic acid in vinegar, constituting the majority of its total acid content. During the aging process, acetic acid serves as a precursor for the formation of esters, ketones, and aldehydes. Additionally, amino acid nitrogen, reducing sugars, and alcohol play crucial roles in this transformation [89,90].

The organic acids in vinegar serve not only as flavor components but also as bioactive compounds. For instance, acetic acid disrupts bacterial outer membranes, inhibits macromolecular synthesis, depletes cellular energy, and increases intracellular osmotic pressure, thereby aiding in the control and treatment of bacterial infections [91]. Mulberry vinegar has been shown to retard carbohydrate digestion, thereby reducing postprandial blood glucose levels and enhancing insulin sensitivity through the inhibition of α-amylase and α-glucosidase activities. The primary constituents of mulberry vinegar are citric acid and tartaric acid [92]. Lactic acid and acetic acid inhibit glucose uptake and fatty acid synthesis by activating AMPK (adenosine monophosphate-activated protein kinase), while promoting fatty acid oxidation. This leads to reduced cholesterol and triglyceride levels, increased high-density lipoprotein cholesterol (HDL-c) content, and inhibited plasma triglyceride synthesis [93]. Acetic acid reduces plasma renin activity, thereby lowering blood pressure. Previous studies have demonstrated that the organic acids in vinegar can ameliorate metabolic disorders and enhance immune function.

#### 4.2.2. Alcohols

Alcohols are generally the most abundant flavor compounds in fruit vinegar. These alcohols are primarily produced through the reduction of aldehydes in fruits, the decarboxylation of amino acids, and the metabolism of the sugars containing aldehyde and ketone groups by yeast. Additionally, a portion is generated by acetobacter [24]. Alcohol is an essential prerequisite for the flavor compounds in fruit vinegar and also serves as one of its key flavor components. As fermentation progresses, some alcohols carried by the raw materials volatilize into the environment, leading to a decrease in their concentration within the vinegar. However, microbial metabolism generates other alcohols with aromatic odors. By comparing the different fermentation stages, it can be inferred that alcohol formation is closely related to microbial activity. Researchers have compared the changes in major flavor components during lemon peel vinegar fermentation and found that yeast fermentation produces para-menth-3-en-1-ol and 1-methyl-4-(1-methylethenyl)cyclohexanol, while acetobacter fermentation yields phenylethyl alcohol [33]. Other researchers have identified 2-ethylhexan-1-ol, octan-1-ol, and benzyl alcohol as unique alcohols in Huyou wine, while cedrol is a distinctive component in Huyou vinegar [25]; propanol, isobutyl alcohol, methionol, and dodecanol are important volatile alcohols in rose wine, in which the concentration of benzyl alcohol has been found to increase [26]. Benzyl alcohol typically exhibits the highest odor activity value among the alcohols in vinegar. It possesses a sweet, rose-like, and honey-like aroma, which is produced through the deamination, hydroxylation, and reduction of phenylalanine [36,94]. Although higher alcohols contribute a pronounced fruity aroma, their sensory quality can be compromised when their concentrations exceed 400 mg/L, leading to undesirable notes of spiciness and bitterness. Conversely, at concentrations equal to or below this threshold, the aroma becomes more complex and nuanced [95]. Therefore, controlling the levels of higher alcohols during the fermentation process is essential for producing high-quality fruit vinegar.

#### 4.2.3. Aldehydes

In vinegar fermentation, aldehydes are typically produced through amino acid degradation or transamination, lipid oxidation, and decomposition. Short-chain aldehydes can also be generated via the Strecker reaction or microbial metabolism, contributing to a pleasant odor. For instance, benzaldehyde, phenylacetaldehyde, and 3-hydroxy-2-butanone impart aromas reminiscent of honey, milk, nuts, and bitter almonds [95]. The presence of carbonyl groups renders the aldehydes chemically reactive, allowing them to be readily reduced to alcohols or oxidized to acids. Consequently, the concentration of aldehydes can serve as an indicator for evaluating quality changes in vinegar. For instance, the benzaldehyde content can be used to assess the degree of vinegar aging [96]. Although the concentration of aldehydes in fruit vinegar is low, they play a crucial role in determining the quality of fruit vinegar due to their low sensory threshold. Phenylacetaldehyde, in particular, is the most prominent aroma component in citrus vinegar [36]. Phenylacetaldehyde is produced through the incomplete oxidation of phenylethanol by acetobacter. Komagataeibacter europaeus enhances the conversion of shikimic acid to aromatic amino acids and their derivatives, such as ferulic acid and phenylacetaldehyde, thereby improving the flavor characteristics of apple vinegar [35]. Nonanal and decanal are the aldehydes with the highest odor activity values in jujube vinegar, serving as key compounds that contribute to its floral and fruity aroma. Furfural, which is formed during the alcohol fermentation stage, imparts a bitter almond aroma and is primarily produced through amino acid deamination and decarboxylation [13]. It is worth noting that the maximum content of aldehydes should be carefully controlled, as excessive levels can impart an odor of spoilage.

#### 4.2.4. Esters

Volatile esters are the primary compounds responsible for the fruity and floral fragrance of fruit vinegar. The formation of ester compounds typically involves microbial enzymatic catalysis, while the non-enzymatic esterification of alcohols and organic acids also contributes significantly to their production. Additionally, certain short-chain fatty acids can be hydrolyzed by yeast to produce esters. Alcohol acetyltransferase plays a crucial role in the formation of acetate esters, initiating the process through the interaction between acetyl-CoA and alcohols [68,95]. The yeast fermentation process can produce a diverse array of esters, although its acetate-producing capability is relatively limited [27,66]. Hanseniaspora exhibits high acetate productivity, generating esters such as isoamyl acetate and isobutyl acetate, which impart banana and strawberry aromas, as well as 2-phenylethyl acetate and hexyl acetate, which contribute rose, honey, fruit, and floral aromas [27]. Metschnikowia pulcherrima exhibits a strong capability for producing ethyl caproate and ethyl caprylate, which impart an apple aroma. Meanwhile, Lachancea thermotolerans produces high yields of ethyl esters that contribute frankincense and fruit flavors [28,97]. Ester compounds exhibit high volatility and a low perception threshold, imparting sweetness, fruity, and floral flavors even at low concentrations [98]. Esters are thus essential for assessing the sensory quality of fruit vinegar. The volatile flavor compounds in aged coconut water vinegar are primarily composed of esters, with ethyl isovalerate and ethyl phenylacetate being the components with the highest odor activity values among the volatile constituents, imparting almond and banana aromas [29]. Acetate esters are primarily formed through the chemical condensation and esterification of acetic acid, which is abundantly produced by acetic acid bacteria. This process contributes significantly to the development of fruity aromas. For instance, isoamyl acetate imparts a banana-like aroma, while 2-phenylethyl acetate confers honey, fruity, and floral notes. Additionally, the concentration of esters can serve as an indicator for the different stages of fruit vinegar fermentation [99]. During the fermentation process of plum vinegar, a significant correlation was observed between acetic acid and several esters, including ethyl acetate, propyl acetate, ethyl pyruvate, ethyl nicotinate, octyl acetate, isoamyl acetate, and ethyl caproate. This correlation can be used to distinguish the different stages of plum vinegar fermentation [24].

### 4.3. Other Components

The degradation of amino acids or the oxidation of unsaturated fatty acids leads to the formation of ketones. The contribution of ketones to the flavor profile of fruit vinegar is generally considered limited, as they are readily oxidized to carboxylic acids due to their chemical instability. This oxidation results in a decrease in ketone content. In hawthorn vinegar, only 3-hydroxy-2-butanone and 2-cyclohexenone were detected, with the former imparting an unpleasant taste characterized by sweet, yoghurt, and dairy notes [30]. Diacetyl, an important flavor component in vinegar, is primarily produced during the alcoholic fermentation stage. It continues to be generated through the decarboxylation of α-acetolactate and 2,3-butanediol by acetobacter species. During the aging stage, diacetyl is further oxidized [31].

The primary types of hydrocarbons include alkanes, olefins, and terpenoids. Due to their high sensory thresholds, hydrocarbons generally contribute minimally to flavor profiles. Alkanes are primarily derived from the decomposition of carotenoids and the auto-oxidation of fats. Olefin compounds, such as d-limonene, α-pinene, β-myrcene, γ-terpinene, and terpinolene, are typically sourced from raw materials. These compounds impart unique orange, pine, and lemon aromas but tend to volatilize progressively during fermentation, leading to a reduction in their concentration [25,33]. Aromatic compounds are derived from the decomposition of aromatic amino acids, which typically produce distinctive aromatic odors. Microorganisms enhance the accumulation of these compounds by activating the metabolic pathways that break down specific aromatic amino acids, including tryptophan, phenylalanine, and tyrosine [68].

Sulfur and nitrogen volatiles are derived from the decomposition of sulfur-containing and nitrogen-containing compounds in cruciferous plants, primarily through the metabolic breakdown pathways of sulfur- and nitrogen-containing amino acids. These compounds are typically present in low relative concentrations or are specific to certain types of fruit vinegar, and they have relatively low flavor perception thresholds. Despite their low concentrations, these substances can significantly influence the flavor profile of fruit vinegar. Notably, dimethyl sulfide, dimethyl disulfide, and dimethyl trisulfide are important volatile components in cabbage vinegar [34]. Sulfides are considered to be the decomposition and thermal degradation products of methylthioproteins, catalyzed by cysteine sulfoxide lyase [100]. Similarly, allyl cyanide and 4-methylthio-butanenitrile can be derived from the degradation of sinigrin and glucoerucin, respectively. These compounds are glucosinolates found in cabbage [101]. Saccharomyces cerevisiae releases volatile sulfides, such as 3-thiohexanol-1-ol, by cleaving the carbon–sulfur bonds of glutathione and cysteine. This process imparts a distinctive thiol flavor to wines [94,95].

## 5. Conclusions and Future Perspectives

Fruit vinegar is a nutritious and flavorful beverage, and moderate consumption can confer numerous health benefits. Flavor is a critical indicator of fruit vinegar quality, significantly influencing consumer preferences, acceptance, and purchasing behavior. The distinctive flavor profile of fruit vinegar primarily results from the metabolic activities of yeast and acetic acid bacteria, which produce a variety of compounds. These flavor components include alcohols, aldehydes, esters, organic acids, and sugars. Alcohols, esters, acids, and aldehydes are primarily produced through amino acid metabolism, fatty acid metabolism, and carbohydrate metabolism. It has been found that the major alcohols in fruit vinegar include ethanol, 1-butanol, phenylethyl alcohol, and 2-methylbutanol. The common esters identified are ethyl acetate, pentyl acetate, phenylethyl acetate, ethyl nicotinate, and propyl acetate. The organic acids in fruit vinegar include crotonic acid, acetic acid, phenylacetic acid, and hexadecanoic acid. Additionally, the sugars and organic acids in fruit vinegar exhibit functional activities such as lowering blood sugar, inhibiting fat absorption, and modulating the immune system. This paper summarizes key research findings on the flavor of fruit vinegar. The current research on microbial metabolism and flavor formation in fruit vinegar primarily focuses on the isolation and screening of dominant microorganisms, a correlation analysis between microbial species and flavor compounds, the identification of volatile flavor substances and key aroma compounds, and the impact of inoculated starters on flavor enhancement and aroma production mechanisms. However, research on the interactions between the microorganisms in fruit vinegar, the interactions between flavor substances, and the regulatory mechanisms of microorganisms and their impact on fruit vinegar flavor remains insufficient. Additionally, further technical research is needed to investigate the effects of raw material diversification, new strains, fermentation methods, and fermentation strategies on the quality of fruit vinegar in practice. The mechanism of flavor formation is complex, but as scientific research deepens, the principles governing flavor components are gradually being elucidated, providing a scientific basis and quality assurance for managing and optimizing flavor during fruit vinegar fermentation.

## Figures and Tables

**Figure 1 microorganisms-13-00477-f001:**
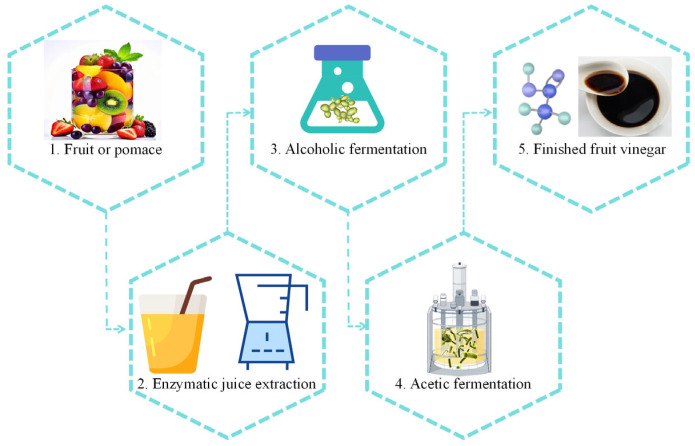
Fruit vinegar liquid fermentation process.

**Figure 2 microorganisms-13-00477-f002:**
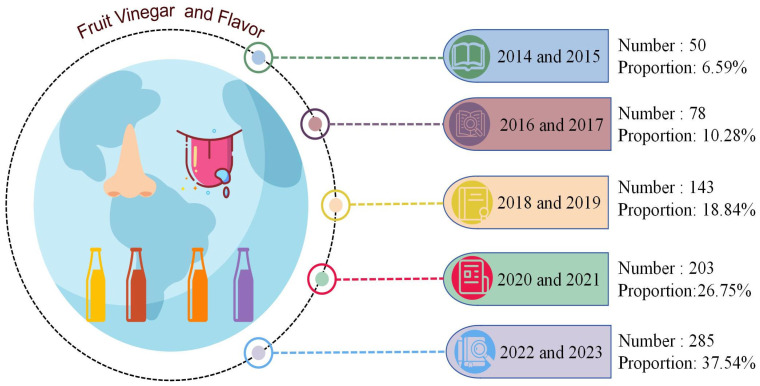
From 2014 to 2023, the number and type of articles published on the Web of Science with the theme of “fruit vinegar” and ”volatile flavor” were analyzed.

**Figure 3 microorganisms-13-00477-f003:**
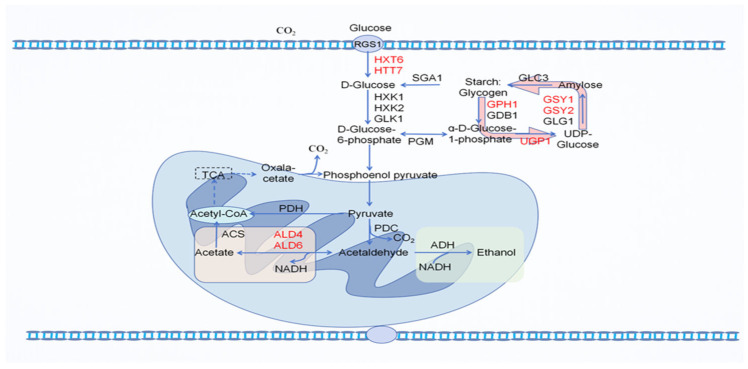
Schematic diagram of carbohydrate metabolic pathway under CO_2_ condition (adapted from [47]). Note: Up-regulated genes are indicated in red.

**Figure 4 microorganisms-13-00477-f004:**
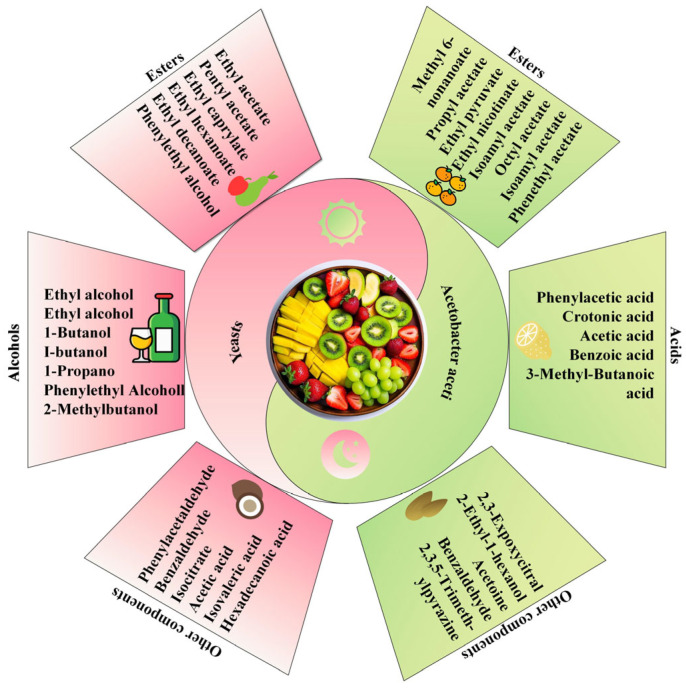
Flavor substances of yeast and acetic acid bacteria in fruit vinegar fermentation.

**Table 1 microorganisms-13-00477-t001:** Characteristics of flavor compounds metabolized by microorganisms.

Type of Strain	Raw Material	Characteristic Flavor Components	Reference
Fruit wines	Apple	1-Butanol; 3-Methyl-Acetate; Acetic acid; Hexyl ester; Acetic acid.	[18]
	Pomegranate	Pentyl acetate; Isopentyl acetate; Ethyl hexanoate; Ethyl caprylate; Isovaleric acid.	[19]
	Cherry	Isopentyl acetate; Hexyl acetate; Ethyl octanoate; Acetic acid; Butanoic acid.	[20]
	Cherry	Benzaldehyde; Linalool; α-Terpineol; Nerol; β-Damascenone.	[20]
	Apple	Phenylacetaldehyde; Ethyl isovalerate; Phenylethyl acetate; Phenylethyl alcohol; Benzaldehyde.	[21]
	Apple	Phenylacetaldehyde; Phenylethyl acetate; Phenylethyl alcohol; Isoamyl acetate; Benzaldehyde.	[21]
	Riesling grapes	Ethyl acetate; 2—Phenyl ethyl acetate; Isoamyl acetate.	[22]
	Riesling grapes	I-butanol; 2-Methylbutanol its acetate ester; 2—methyl butyl acetate.	[22]
	Persimmon	Methyl acetate; Ethyl lactate; Betaine; Aconitic acid; Acetoin; 2,3-Butanediol; Isocitrate.	[23]
	Prune	2,3-Butanediol; 2-Methylbutanol; Ethyl heptanoate; Ethyl pyruvate; Ethyl acetate	[24]
	Huyou (Citrus changshanensis)	2-Phenylethanol; 3-Methyl-1-butanol; Ethyl decanoate; Diethyl; Ketones; Aldehydes; Heterocyclic.	[25]
	Rosehip fruit (*Rosa canina* L.)	3-Methyl-1-butanol; Phenylethyl alcohol; Octanoic acid; Ethyl acetate; Acetic acid; Ethyl decanoate; Ethyl lactate; Vitispirane; 1-P-menthen-9-al; Ethyl caprylate; Dodecanol; Propanol.	[26]
	Red grapes (Tinta Roriz)	1-Propanol; 2-Phenylethyl acetate; 3-(Methylthio)Propionic acid.	[27]
	Riesling grapes	Isoamyl acetate; 2-Methyl butyl acetate; 2-Phenyl-ethyl acetate.	[28]
	Coconut water	2-Methylbutanol; Ethyl octanoate; Ethyl hexanoate; Isoamyl acetate.	[29]
	Hawthorn (Crataegus tanacetifolia)	Ethanol; 2-Phenethyl alcohol; 3-Methyl-1-butanol; Ethyl acetate; Acetic acid 2-phenylethyl ester; 2-Methyl-1-propanol; 2,4-Dimethylbenzaldehyde; Hexadecanoic acid; Ethyl octanoate; Isovaleric acid.	[30]
	Pineapples (Ananas comosus)	2-Methyl-propyl acetate; 2-Phenylethyl acetate; 3-Methyl butanol acetate; Ethyl acetate; Ethyl hexanoate; Ethyl octanoate; 2-Phenyl ethano; 3-Methyl butanol.	[31]
Fruit vinegars	Black tea	(Z,E)-farnesol; Linalyl formate; 3-Hydroxybutyrate ethyl; α-Santalol; 2,3-Expoxycitral; α-Bisabolol oxide B; Ionene; Dihydroactinidiolide.	[3]
	Kaki (persimmon)	1-(3H)-Isobenzofuranone; 2-(3H)-Furanone; 5-Dodecyldihydro-	[32]
	Lemon Peel	6-nonynoic acid; methyl ester; 2,5-Octadecadiynoic acid; Methyl ester; Bicyclo [3.1.1]hept-2-en-4-ol; 2,6,6-trimethyl.	[33]
	Cabbage	Dimethyl sulfide; Dimethyl disulfide; Dimethyl trisulfide allyl cyanide; 4-Methylthio-butanenitrile; 3-Hexene-1-ol; Crotonic acid.	[34]
	Prune	Linalool; Thyl acetate; Propyl acetate; Ethyl pyruvate; Ethyl nicotinate; Octyl acetate; Isoamyl acetate; Ethyl caproate; Acetic acid.	[24]
	Huyou (Citrus changshanensis)	Ethyl acetate; Methyl acetate; Hexyl acetate; Phenethyl acetate; γ-Nonalactone; 2,4,5-Trimethyl-1,3-oxazole; 2,3,5-Trimethylpyrazine.	[25]
	Apple	Hexanoic acid ethyl ester; Phenethyl acetate; Isoamyl acetate.	[35]
	Citrus	Acetic acid; Phenethyl acetate; Phenethyl alcohol; Benzaldehyde.	[36]
	Coconut water	Phenylethyl acetate; Ethyl hexanoate; Isoamyl acetate; Benzaldehyde.	[29]
	Pineapples (Ananas comosus)	2-Ethyl-1-hexanol; 3-Methyl-butanoic acid; Acetoine.	[31]

## Data Availability

The original contributions presented in this study are included in the article, further inquiries can be directed to the corresponding author.

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
