# Peer review of "The Metabolic Pathways of Yeast and Acetic Acid Bacteria During Fruit Vinegar Fermentation and Their Influence on Flavor Development"

_microorganisms, 2025, doi:10.3390/microorganisms13030477_

Round 1
Reviewer 1 Report
Comments and Suggestions for Authors
General issues:
The article is poorly constructed. The introduction is too long, not insightful and superficial. Numerous repetitions and truisms make the content difficult to understand. Interesting content doesn't start until Chapter 4.
Detailed issues:
Table 1 – what is the key to read the table? Is it alphabetic order by microorganism? By raw material? In order to year of publication? Neither the choice of articles, neither their order in the table is clear.
Line 220 – alcohols can also enhance the sweetness and alcohol content of wine [55]. – Phrase is not understandable and the cited article describes beer production.
Lines 230-235 – totally not understandable. The paragraph should be rewritten.
In this chapter there is not clearly defined what factors positively affects the taste of vinegar. Is crotonic acid a positive or negative ingredient?
Is the reference to a publication about taste of wine justified? The content of alcohol and acetic acid in wine and their content in vinegar have completely opposite significance for the perception of taste.
Lines 237-243 – Very poorly written paragraph. There is three times repeated that acetic acid bacteria produces acetic acid.
Lines 239-240 – The higher the concentration of acetic acid, the higher the acidity of fruit vinegar. – Isn’t it obvious?
Line 244 – These acetic acid bacteria exhibit strong enzyme activity. – It’s quite typical for bacteria to exhibit strong enzyme activity.
Lines 245-246 – It can not only convert ethanol and glucose into acetic acid and gluconic acid, but also oxidize other alcohols and sugars to generate corresponding acids - corresponding to what? Not described properly.
Line 280 – heteropolysaccharides, such as acetyl or xylans and glucans. – not understandable. Is acetyl a saccharide?
Does the whole chapter about polysaccharides refers in any way to vinegar taste either the efficiency of vinegar production?
Line 328 – the crude polysaccharides ( arabinose, mannose, galactose, rhamnose and galacturonic acid ) – factual error, these are monosaccharides not polysaccharides.
Line 363 – conclusions are unjustified and too far going. They refer to amino acids in overall not the vinegars containing amino acids.
Line 392 – The main component is citric acid [81] – is that true?
Comments on the Quality of English Language
Punctuation and language need drastic improvement.
Reviewer 2 Report
Comments and Suggestions for Authors
GENERAL COMMENTS
Dear editor, dear authors,
I have carefully read the paper entitled “The Metabolic Pathways of Yeast and Acetic Acid Bacteria in the Process of Fruit Vinegar Fermentation and Their Impacts on the Flavor of Fruit Vinegar”. This paper reviews the roles of yeast and acetic acid bacteria during fruit vinegar fermentation, their metabolic pathways, the metabolites they produce and their influence on the flavor profile, as well as the benefits they could produce for human health. The review is well written, easy to follow and compact. To my knowledge, there aren’t many review papers that cover the topic of relation between yeast and acetic acid bacteria metabolism in fruit vinegar fermentation and the flavor components which makes this paper a good addition to the field.
My suggestion is that the paper should be accepted after some minor revisions listed below.
SPECIFIC COMMENTS
1. In some parts of the manuscript, the names of bacterial genera and species are not written in italics. For example, lines 62, 63, 125, 132, 150, 166, 167, 251, 325 etc. The authors should check the whole manuscript and correct this where needed.
2. The second2: Word increasing is used twice here, there is no need for it, the second one could be omitted. For example, “…the number of articles was
increasing year by year, by 5 times.” or similar.
3. Lines 240-243: “The main strain used in the production of vinegar and fruit vinegar beverages is acetic acid bacteria, which are divided into acetic acid bacteria and gluconic acid bacteria, among which acetic acid bacteria are mainly acetic acid bacteria.” This sentence should be reformulated as it is a bit confusing.
4. Figure 4 is quite small which makes the text in it hard to read, it should be enhanced for the text to be more readable.
5. Lines 309 to 316: These two paragraphs seem to have been left over from the template; they should be deleted.
6. Lines 318, 338, 367, 447: These subtitles should be transferred to plural, since more of them are mentioned. Organic Acids, Amino Acids, Esters, etc.

Round 2
Reviewer 1 Report
Comments and Suggestions for Authors
The authors introduced the necessary changes to the manuscript. The quality of language and clarity of the article structure have been significantly improved.